# Which priors matter?
# Benchmarking models for learning latent dynamics

**Aleksandar Botev**
DeepMind
London
botev@deepmind.com

**Andrew Jaegle**
DeepMind
London
drewjaegle@deepmind.com

**Peter Wirnsberger**
DeepMind
London
pewi@deepmind.com

**Daniel Hennes**
DeepMind
London
hennes@google.com

**Irina Higgins**
DeepMind
London
irinah@deepmind.com

## Abstract

Learning dynamics is at the heart of many important applications of machine learning (ML), such as robotics and autonomous driving. In these settings, ML algorithms typically need to reason about a physical system using high dimensional observations, such as images, without access to the underlying state. Recently, several methods have proposed to integrate priors from classical mechanics into ML models to address the challenge of physical reasoning from images. In this work, we take a sober look at the current capabilities of these models. To this end, we introduce a suite consisting of 17 datasets with visual observations based on physical systems exhibiting a wide range of dynamics. We conduct a thorough and detailed comparison of the major classes of physically inspired methods alongside several strong baselines. While models that incorporate physical priors can often learn latent spaces with desirable properties, our results demonstrate that these methods fail to significantly improve upon standard techniques. Nonetheless, we find that the use of continuous and time-reversible dynamics benefits models of all classes.

## 1 Introduction

Much progress across deep learning has been driven by the formulation of good *inductive biases*, i.e. by exploiting known properties of a data domain (such as its symmetries) to design architectures for data-efficient learning. This principle has informed the design of widely adopted building blocks such as convolutional neural networks [1, 2] (which exploit the translational invariance of many natural signals, like images), recurrent neural networks [3] (which similarly exploit time-translational invariance), and graph-structured networks like transformers [4, 5] (which exploit the permutation invariance of graph-structured latent variables).

Many in the machine learning community have expressed the need for better ways to encode inductive biases based on physical laws into neural network architectures [6, 7]. Indeed, if you want to understand how the world will change, simple physical principles like the laws of motion are a good place to

start. However, the question of what design principles should be incorporated directly into the model architecture and what should be learned from data still remains open [8]. For example, while classical robotics systems often incorporate the laws of mechanics into their planning modules to exploit basic physics at deployment, the exact method for doing so successfully depends on the application (see, e.g. [9], Ch. 13 and the literature on task-and-motion planning [10]).

Although the dynamics of systems depicted in real-world videos are governed by the laws of physics, we do not always have access to a good model of these dynamics. Rather than attempting to model each video analytically, a more general solution may be to learn to capture the data's latent dynamics using an architecture with an appropriate inductive bias. This intuition has been referred to as the Hamiltonian manifold hypothesis [11], which conjectures that natural images lie on a low-dimensional manifold embedded within a high-dimensional pixel space and *natural sequences of images trace out paths on this manifold that follow the equations of an abstract Hamiltonian*. The goal of this paper is to characterize the current state of inductive biases inspired by classical mechanics and to ask whether models that use these inductive biases have the potential to serve as general tools for learning dynamics from high-dimensional observations.

To this end, we introduce a test suite of 17 challenging datasets governed by known equations of motion that evaluate the models on their ability to capture different kinds of dynamics, from toy physics problems, to molecular dynamics, to learning in zero-sum games or movements in 3D environments. We use this test suite to rigorously compare the most widely adopted physics-inspired models and a set of strong, standard baselines in a common framework. We conduct a thorough review of the existing literature, extracting key ideas for incorporating priors from classical mechanics into deep neural networks. In particular, we investigate the role of Hamiltonian vs Lagrangian inductive biases, modelling dynamics as continuous vs discrete, and training models to predict forward and/or backward in time. We test the models on their ability to predict the state of the system significantly further in time than what they are trained on. Our results demonstrate that the most important inductive biases for learning dynamics from high-dimensional observations are continuity and time-reversibility. In summary, our contributions are as follows:

1. We introduce 17 new datasets of dynamical systems of various complexity, spanning real physical systems, learning in cyclic games, and camera movements in a 3D room

2. We implement several recently proposed models with physical priors and perform a detailed evaluation, including comparisons to strong baselines that make few assumptions about the dynamics

3. We open source the datasets, which are available for download on `https://console.cloud.google.com/storage/browser/dm-hamiltonian-dynamics-suite` [1]

## 2   Related Work

Recently a large number of papers have introduced inductive biases inspired by either the Hamiltonian or the Lagrangian formulation of classical mechanics to deep neural networks. However, most of the models are never compared against each other, and each paper tends to re-implement its own datasets and evaluation protocols. Hence, it is not clear whether any of these models can be seen as generally useful for learning dynamics from images, with the view of using them in applications such as robotics or reinforcement learning in natural 3D environments.

In this section we give a short overview of the Hamiltonian and Lagrangian formalisms of classical mechanics, before systematising the related literature in order to extract the key themes for re-implementation.

### 2.1   Hamiltonian and Lagrangian mechanics

Hamiltonian and Lagrangian mechanics are two different mathematical reformulations of Newton's equations of motion for describing energy-conservative dynamics [12]. Because such systems conserve energy, it is possible to predict the state of the system over significantly more steps forward

---

[1]The code for generating the datasets is available on `https://github.com/deepmind/dm_hamiltonian_dynamics_suite` and for reproducing all experimental results on `https://github.com/deepmind/deepmind-research/tree/master/physics_inspired_models`.

or backwards in time than was used during training, making this an attractive bias to build into deep neural networks. Many important kinds of non-physical dynamics can also be modelled using this formulation, including those of GAN optimisation [13–16], multi-agent learning in zero-sum games [17–19], or the transformations induced by flows in generative models [20]. Both formalisms describe the exact same dynamics, but in a different coordinate frame – one uses the position $\boldsymbol{q}$ and momentum $\boldsymbol{p}$, while the other uses position $\boldsymbol{q}$ and velocity $\dot{\boldsymbol{q}}$. As such one can "translate" between Hamiltonian and Lagrangian mechanics without any loss of generality.

**Hamiltonian mechanics** The Hamiltonian formalism describes the continuous time evolution of a system in an abstract phase space, with state $\boldsymbol{s} = (\boldsymbol{q}, \boldsymbol{p}) \in \mathbb{R}^{2n}$, where $\boldsymbol{q} \in \mathbb{R}^n$ is a vector of position coordinates, and $\boldsymbol{p} \in \mathbb{R}^n$ is the corresponding vector of momenta. The Hamiltonian function $\mathcal{H} : \mathbb{R}^{2n} \to \mathbb{R}$ is the fundamental object of interest that maps the state $\boldsymbol{s}$ to a scalar representing the total energy of the system. It can be expressed as the sum of the kinetic energy $T$ and potential energy $V$. The Hamiltonian specifies a vector field over the phase space that describes all possible dynamics of the system, whereby only a single trajectory passes through every point in the phase space. In the models we consider energies that are time independent and separable, meaning that the potential energy depends only on the position and the kinetic energy depends only on the momentum - $\mathcal{H}(\boldsymbol{q}, \boldsymbol{p}) = V(\boldsymbol{q}) + T(\boldsymbol{p})$. This choice allows us to use symplectic numerical integrators for simulating the system, which are considered to produce more accurate long term behaviour than standard integrators, and is the standard in the literature. The time evolution of the system is given by the Hamilton's equations of motion:

$$\frac{\mathrm{d}\boldsymbol{q}}{\mathrm{d}t} = \frac{\partial \mathcal{H}}{\partial \boldsymbol{p}}, \quad \frac{\mathrm{d}\boldsymbol{p}}{\mathrm{d}t} = -\frac{\partial \mathcal{H}}{\partial \boldsymbol{q}}. \tag{1}$$

This expression makes energy conservation obvious, since $\frac{\mathrm{d}\mathcal{H}}{\mathrm{d}t} = \frac{\partial \mathcal{H}}{\partial \boldsymbol{q}} \frac{\mathrm{d}\boldsymbol{q}}{\mathrm{d}t} + \frac{\partial \mathcal{H}}{\partial \boldsymbol{p}} \frac{\mathrm{d}\boldsymbol{p}}{\mathrm{d}t} = 0$.

**Lagrangian mechanics** The Lagrangian formalism describes the continuous time evolution of a system in an abstract state-space, with state $\boldsymbol{s} = (\boldsymbol{q}, \dot{\boldsymbol{q}}) \in \mathbb{R}^{2n}$, where $\boldsymbol{q} \in \mathbb{R}^n$ is a vector of position coordinates, and $\dot{\boldsymbol{q}} \in \mathbb{R}^n$ is its time derivative or a vector of velocities. The Lagrangian function $\mathcal{L} : \mathbb{R}^{2n} \to \mathbb{R}$ is the fundamental object of interest which maps the state $\boldsymbol{s}$ to a scalar representing the immediate change in the *action* of the system. We only consider time-independent energies in this work and assume that the potential energy $V$ depends only on the positions, while the kinetic energy $T$ can depend on both state variables. Although the functional form of the kinetic energy breaks with our assumption of separable energies in the Hamiltonian section, we adopt it here as it was found beneficial for Lagrangian models (Zhong and Leonard [21], Allen-Blanchette et al. [22]). Given the Lagrangian $\mathcal{L}(\boldsymbol{q}, \dot{\boldsymbol{q}}) = T(\boldsymbol{q}, \dot{\boldsymbol{q}}) - V(\boldsymbol{q})$ we can then derive the equations of motion from the principle of least action, and write them as

$$\frac{\mathrm{d}\boldsymbol{q}}{\mathrm{d}t} = \dot{\boldsymbol{q}}, \quad \frac{\mathrm{d}\dot{\boldsymbol{q}}}{\mathrm{d}t} = \left( \frac{\partial^2 \mathcal{L}}{\partial \dot{\boldsymbol{q}}^2} \right)^{-1} \left( \frac{\partial \mathcal{L}}{\partial \boldsymbol{q}} - \frac{\partial^2 \mathcal{L}}{\partial \dot{\boldsymbol{q}} \partial \boldsymbol{q}} \dot{\boldsymbol{q}} \right). \tag{2}$$

### 2.2 Systematising models with physical priors

**Learning from state, observation or time derivative** Most models that incorporate Hamiltonian or Lagrangian priors into their learning use ground truth state information as input (see 23 for a review). The advantage of these models is that they only need to infer the Hamiltonian or the Lagrangian, without having to learn in addition a state representation. This makes also the evaluation easier as we only need to calculate the distance between the ground truth states and the states predicted by the model across the trajectory. On the other hand, such models may not be applicable to problems for which ground truth data is not available or to pixel-based observations.

To address this issue, a number of approaches have been proposed that augment their physics-inspired models of dynamics with encoder/decoder modules for inferring the low-dimensional states from high-dimensional pixel observations [11, 24, 25, 22, 21]. While these methods have the promise of being applicable to a broader class of problems that require modelling dynamics with physical priors, they have so far only been evaluated by measuring the quality of reconstructions in the observation space, which may not be a good measure to assess the quality of the learnt state space or dynamics. In this paper we are primarily interested in this class of models that learn latent dynamics from high-dimensional observations.

Finally, while some models predict the state or the observation directly at each time step, others predict their time derivatives instead, e.g. Greydanus et al. [26], Choudhary et al. [24]. The latter can be

problematic, because such time derivatives are often not available in the original data and have to be estimated with finite differences, thus potentially introducing a source of error in the training data.

**Using the Hamiltonian or the Lagrangian prior**  Although both formalisms describe equivalent dynamics, they represent them in different coordinate systems. Cranmer et al. [27] and Lutter et al. [28] argue that the Hamiltonian formulation could be more challenging for some problems because of the requirement that positions and momenta be canonical coordinates. On the other hand, the Lagrangian formulation requires calculating second-order derivatives, which can be computationally expensive and can lead to instabilities in model training. Zhong et al. [23] report comparable results with both formulations for training on state space data, however no such comparison exists when it comes to learning from high dimensional observations. Therefore, it is still an open question which, if any, of the two formalisms is better for learning in this latter setting.

**Structured or unstructured priors**  Priors can be incorporated into the model in various ways. Some models make assumptions on the type of dynamics to be modelled (e.g. rigid body), which in turn specifies the form of the Hamiltonian, and hence allows us to use neural networks to estimate particular parts of the function while incorporating a lot of structure from its analytical form (e.g. 29, 30, 28, 21). Other approaches assume that the Hamiltonian is separable, i.e. that kinetic and potential energies depend on different parts of the state (e.g. 31, 11). Unlike the rigid body assumption, this separability assumption is fairly general and holds for many interesting problems. It allows us to use symplectic integrators which naturally conserve energy. However, it may also be possible to avoid any assumptions on the nature of the Hamiltonian and instead model it as a single function (e.g. 32, 27, 33).

**Using single or multiple step predictions**  During training some models predict a single time step (e.g. 26, 27) while others perform multi-step predictions (e.g. 34, 22, 11). It is not clear whether one way or the other is better.

**Explicit time reversal**  Due to energy conservation, it should be possible to reverse the dynamics of models that have faithfully captured the underlying system without any loss in prediction quality. Some authors however directly train their models to predict both forward and backwards in time (e.g. 31).

**Integrator choice**  Since the models reviewed in this section all use continuous dynamics, the choice of integrator used inside the model becomes important. While many approaches use the simple first-order Euler integrators (e.g. 26), others use higher-order integrators from the Runge-Kutta family (e.g. RK2, RK4). Neither RK4 nor Euler integration is guaranteed to preserve the energy of the system over long integration windows, and in practice both will produce estimates that drift away from the true system dynamics over timescales that are relevant for simulating real systems. Hence, some approaches use the symplectic leap-frog integrator (e.g. 35, 11, 31), which is guaranteed to preserve the special form of the Hamiltonian, if it is separable, even after repeated application.

## 3   Models

The main focus of this paper is to understand whether models that incorporate physical priors are good general alternatives to RNNs for learning dynamics from high-dimensional observations. To reduce the plethora of architectural choices we restrict our investigation to models which are Markovian and which treat the latent representation as a single vector without making any further assumptions or introducing further inductive biases (for example some previous works treat the latent representation as a spatial image or a graph [11, 36, 37, 4]). To this end, we implemented the most general versions of the Hamiltonian and Lagrangian-based models synthesised through our literature review. Furthermore, in order to compare these models to the baselines as fairly as possible, we reuse the same model architecture in all of the evaluated models, only changing the component that learns the dynamics of the system (see Figure 1). In this section, we first describe the general model architecture that is common across all models, before going into the details of the different dynamics modules.

### 3.1   Variational Autoencoder architecture

We use a Variational Autoencoder (VAE) [38, 39] as the basis for our models, inspired by the Hamiltonian Generative Network (HGN) [11]. It consists of the encoder network that learns to embed

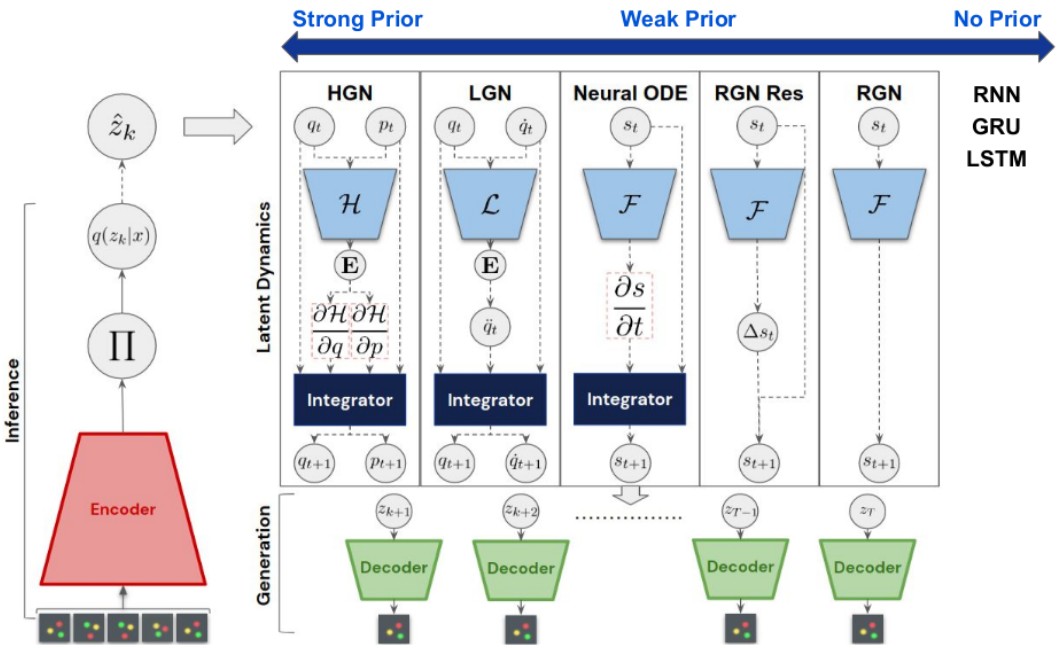

Figure 1: Schematic of all generative models compared.

a sequence of images $(\boldsymbol{x}_0, ... \boldsymbol{x}_T)$ to a lower-dimensional abstract state $\boldsymbol{s}_T \sim q_\phi(\cdot|\boldsymbol{x}_0, ... \boldsymbol{x}_T)$. The dynamics are modelled in this inferred state space deterministically, which makes it sufficient to estimate the latent state only at the first time step. Finally, the decoder network maps the position coordinates of the state space back to the image space $p_\theta(\boldsymbol{x}_t) = d_\theta(\boldsymbol{q}_t)$ (see Fig. 1 for a schematic). The model is trained using the $\beta$-VAE objective [40], since varying the $\beta$ hyperparameter we can switch between training in the deterministic or the stochastic regime:

$$\mathcal{L}(\theta) = \mathbb{E}_{p(\boldsymbol{x})}\left[\mathbb{E}_{q_\phi(\boldsymbol{s}_k|X)}\left[\frac{1}{T+1}\sum_{t=0}^{T}\log p_\theta(\boldsymbol{x}_t|\boldsymbol{s}_t)\right] - \beta \mathcal{D}_{KL}(q_\phi(\boldsymbol{s}_k|X)|p(\boldsymbol{s}_k))\right], \qquad (3)$$

where $q_\phi(\boldsymbol{s}_k|X)$ is the approximate distribution output by the encoder, $p(\boldsymbol{s})$ is the latent prior and $\log p_\theta(\boldsymbol{x}|\boldsymbol{s})$ is the distribution from the decoder. The encoder and decoder networks are implemented as convolutional modules with leaky relu activations similar to the original HGN paper. All dynamics modules use swish activations and are implemented as a four-layer MLP with hidden layer size of 250. The generative models are trained by inferring the latent state from the first 5 images, and reconstructing 60 images after. Hence, during training every model is presented with sequences of length 65 sampled randomly from the longer trajectories included in the datasets.

### 3.2 Hamiltonian and Lagrangian dynamics modules

We implement the dynamics module with Hamiltonian (HGN) and Lagrangian (LGN) priors by using two neural networks to parameterise the kinetic energy $T$ and the potential energy $V$. Given the latent state $\boldsymbol{s}$ produced by the encoder and the two networks that compute the kinetic and potential energy we can simulate the equations of motion as described in Section 2 using any standard numerical integrator. For the HGN model we use the leap-frog integrator. For the LGN model we instead use the adaptive stepsize (Dormand-Prince) Runge-Kutta method included in JAX [41].

One challenge with implementing the LGN was that its equations of motion depend on the inverse of the Hessian of the Lagrangian with respect to the velocity variables, which we found both computationally expensive and numerically unstable to compute. Following Allen-Blanchette et al. [22], we therefore defined the kinetic energy for this specific model as $\frac{1}{2}\dot{\boldsymbol{q}}^T\left(\mathcal{F}(\boldsymbol{q})\mathcal{F}(\boldsymbol{q})^T + \lambda I\right)\dot{\boldsymbol{q}}$, where the network $\mathcal{F}$ outputs a lower-triangular matrix.

### 3.3 Neural ODE

In order to provide a baseline for learning continuous time dynamics without physical priors, we implemented a Neural ODE model [42]. Unlike HGN and LGN that use MLPs to parametrise the functions that give rise to the time derivatives of the latent state, here the time evolution of the latent state is directly parameterised by an MLP: $\dot{s} = \mathcal{F}(s)$ (see Figure 1). The network architecture used for $\mathcal{F}$ is equivalent to that used for the kinetic and potential networks in the HGN, with the only difference being that it outputs not a scalar, but a vector of the same dimensionality as the input. Since Neural ODE has no bias towards learning time-reversible dynamics, we used two versions of the model: one trained only forward in time, and another one trained both forward and backward in time following Huh et al. [31] and abbreviated as TR. For integrating the ODE we used either an explicit RK2 method, which allows us to differentiate through the integrator during training, or the adapative stepsize method included in JAX.

### 3.4 Recurrent Generative Network

In order to investigate the importance of modelling the dynamics as continuous instead of discrete, we also implemented a Recurrent Generative Network (RGN) that uses the same MLP as the Neural ODE model to parametrise the discrete state evolution in latent space. For this model, the time evolution is given by $s_{t+\Delta t} = \mathcal{F}(s_t)$. In addition, we evaluated a residual version of the RGN, for which the output of the network is the update to the previous state, i.e. $s_{t+\Delta t} = s_t + \mathcal{F}(s_t)$ (see Figure 1, RGN Res). This can be seen as a discrete version of the Neural ODE.

### 3.5 Autoregressive models

The most popular machine learning models for modelling sequences are Recurrent Neural Networks (RNN) and their variants, the Long-Short Term Memory networks (LSTM) [43] and the Gated Recurrent Unit networks (GRU) [44]. Hence, we also include these models as baselines. They use the same network architecture for mapping between their hidden state and observation as the encoder and decoder networks of the generative models. In order to make them of "similar capacity" as the other models, we use four recurrent layers in the AR baselines to match the number of layers in the MLPs used to model dynamics in the other models.

## 4  Datasets

Our goal was to develop a benchmark of datasets, with varying complexity of the simulated dynamics and visual richness, which could serve as a measure of progress for current and future models incorporating physical priors from classical mechanics. To this end, we developed a suite of 17 datasets (see Figure 2) ranging from simple physics problems (Toy Physics), including mass-spring and pendulum datasets which are meant to be learnable by the existing methods, to

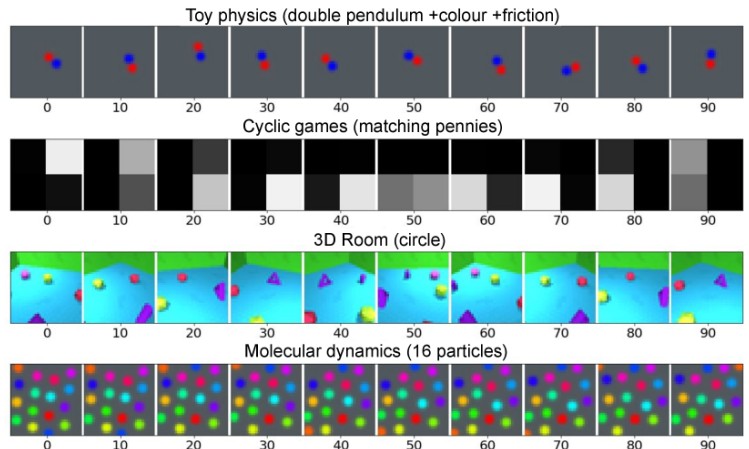

Figure 2: A visual depiction of several of the datasets introduced.

more challenging versions of these, like Toy Physics with friction or colour modifications, to more realistic datasets of molecular dynamics (MD). We included datasets of learning dynamics in nontransitive zero-sum games (Cyclic-games) as examples of Hamiltonian dynamics that have high-dimensional observations that are not pixel images, and which may be important to learn due to their relation to GAN optimisation. Finally, we include a dataset of motion in 3D simulated environments (3D Room), designed specifically to resemble a standard RL environment, where the dynamics may be well-described by a Hamiltonian, but may be hard to infer due to occlusion and cropping.

Emphasising again that the main goal of this paper is to investigate how well models with different physics-inspired priors can learn dynamics from high dimensional observations, we constrained our datasets to systems which can easily be represented as images, and which should be feasible for the current models to learn. For each dataset we created 50k training trajectories, and 20k test trajectories, with each trajectory including image observations, ground truth phase space used to generate the data, and the first time derivative of the ground truth state to facilitate the use of these datasets on models that require this information (e.g. 26).

## 4.1 Toy Physics

We simulate four toy physics systems with various degrees of complexity of the underlying Hamiltonians (see Table 2): mass-spring, pendulum, double-pendulum and 2-body system. While these systems have been used in previous work (e.g. 26, 11), no canonical implementation of these datasets exist, and we have found that the performance of the models can vary significantly based on the particular implementation choices.

The first dataset, referred to as Mass Spring, describes simple harmonic motion of a particle attached to a spring. Each trajectory is generated with a fixed spring force coefficient $k = 2$, and the mass of the particle $m = 0.5$. The Pendulum dataset represents the evolution of a particle attached to a pivot, such that it can move freely. The system is simulated in angle space, such that it is one dimensional. Each trajectory is generated with a fixed mass of the particle $m = 0.5$, gravitational constant $g = 3$ and pivot length $l = 1$. A similar, but significantly more difficult system, is the Double Pendulum, which is a system like the previous, but with a second pendulum attached to the first one. This leads to significantly more complicated dynamics due to the coupling between the two particles [45]. This dataset hyperparameters are the same as in the Pendulum. The final dataset that we consider is the Two Body Problem, which describes the gravitational motion of two particles in the plane. The hyperparameters of this dataset are the masses of each individual particle $m_i = 1$ and the gravitational constant $g = 1$. For each system we simulate the corresponding dynamics, with the numerical integrator provided by 'scipy' [46], starting from random initial conditions. We sample the system state at every $\Delta t = 0.05$ for observations. Since all of the systems can be represented as particles in a 2D plane we render each particle as a circle with area proportional to its mass.

**+colour**    In order to test how well models can cope with additional variations in the observations, we also generate additional versions of the four toy physics datasets, where the particles are rendered in random colours and in random positions. These datasetes are abbreviated with '+c'. The main goal is to explore whether models can learn the "colour" and "translation" invariances. Additionally, for these datasets we sample randomly the hyperparameters of the Hamiltonians, like the mass of the particle in the mass-spring system. The exact sampling procedures for the hyperparameters of each '+c' dataset and other simulation details are listed in Appendix A.1.1.

**+friction**    Finally, for Mass Spring, Pendulum and Double Pendulum, we also construct a non-energy conserving dataset. This is achieved by introducing a friction coefficient $\lambda$, which dampens the updates to the momentum. The underlying dynamical system in this case is modified from the classical Hamilton's equations by modifying the time derivative of the momenta to $\frac{d\boldsymbol{p}}{dt} = -\frac{\partial \mathcal{H}}{\partial \boldsymbol{q}} - \lambda \frac{\partial \mathcal{H}}{\partial \boldsymbol{p}}$. The introduced friction term decreases the overall energy of the system proportionally to the squared velocity according to $\dot{\mathcal{H}} = -\lambda \dot{q}^2$. We abbreviate the datasets including friction with '+f'.

## 4.2 Multi-agent cyclic games

Non-transitive zero-sum games can induce cyclic dynamics, as observed in evolutionary dynamics, GAN optimization, and multi-agent learning [47, 17, 13–16, 18, 19]. Here we consider two prominent examples of such games: matching pennies and rock-paper-scissors.

We use the well-known continuous-time multi-population replicator dynamics to drive the learning process. Replicator dynamics [48] are defined by a system of differential equations that couple the relative size of each population with the relative fitness of that population. Replicator dynamics can be used to model the dynamics of multi-agent systems by assigning one population to each possible strategy in the game: intuitively, the more often a given strategy wins (i.e. the fitter that population) the more often it will be played. Replicator dynamics have a long-standing connection to evolutionary game theory and multi-agent learning [49].

In the case of the two-population games we consider here, the continuous-time replicator dynamics are given by:

$$\dot{x}_i = x_i \big[ (\boldsymbol{A}\boldsymbol{y})_i - \boldsymbol{x}^T \boldsymbol{A}\boldsymbol{y} \big]$$
$$\dot{y}_j = y_j \Big[ \big(\boldsymbol{x}^T \boldsymbol{B}\big)_j - \boldsymbol{x}^T \boldsymbol{B}\boldsymbol{y} \Big], \tag{4}$$

where $\boldsymbol{A} = -\boldsymbol{B}$ are the payoff matrices of a zero-sum game for the row and column player respectively, i.e. matrices that indicate whether the strategy of the row or column player wins. $(\boldsymbol{x}, \boldsymbol{y})$ is the joint strategy profile, i.e. the probability of choosing each strategy for each of the two populations.

For more details of the generation protocol for this dataset see Appendix A.1.2. As all other datasets use images as inputs, we define the observation as the outer product of the strategy profiles of the two players: $\boldsymbol{x} \otimes \boldsymbol{y}$. The resulting matrix captures the probability mass that falls on each pure joint strategy profile (i.e. each joint action). In this dataset, the observations are a loss-less representation of the ground-truth state.

## 4.3 Molecular Dynamics

The goal of this set of datasets is to benchmark the performance of our models on problems involving complex many-body interactions. To this end, we consider an interaction potential commonly studied using computer simulation techniques, such as molecular dynamics (MD) or Monte Carlo simulations. In particular, we generated multiple datasets employing a Lennard-Jones (LJ) potential [50], which is a popular benchmark problem and an integral part of more complex force fields used, for example, to model water [51] or proteins [52].

We generated two LJ datasets of increasing complexity to benchmark the models: one comprising only 4 particles at a very low density and another one for a 16-particle liquid at a higher density. We note that the system sizes considered here are much smaller than the problems typically studied using MD. Moreover, the LJ potential has been studied extensively in the physics literature in both two and three spatial dimensions, with the latter being more common. However, for the purpose of this comparison, we find a small, two-dimensional system well suited owing to its ease of illustration. A summary of the LJ potential, the thermodynamic states we chose for comparison, and details on the simulation protocol are provided in Appendix A.1.3.

For rendering these MD datasets we use the same scheme as for the Toy Physics datasets. All masses are set to unity and we represent particles by circles of equal size with a radius value adjusted to fit the canvas well. The illustrations are therefore not representative of the density of the system. In addition, we assigned different colours to the particles to facilitate tracking their trajectories.

## 4.4 3D Room

To evaluate the ability of models to deal with complex 3D visuals, we generated a dataset of MuJoCo [53] scenes consisting of a camera moving around a room with 5 randomly placed objects. The objects were sampled from four shape types: a sphere, a capsule, a cylinder and a box. Each room was different due to the randomly sampled colours of the wall, floor and objects similar to Kabra et al. [54]. The dynamics were created by rotating the camera either around a single randomly sampled parallel of the unit hemisphere centered around the center of the room, or on a spiral moving down the unit hemisphere. In particular, for each trajectory we sample an initial radius and angle, which we then convert into the Cartesian coordinates of the camera. The dynamics are discretised by moving the camera using step size of $1/10$ degrees in a way that keeps the camera on the unit hemisphere while facing the centre of the room. For the 3D Room Spiral dataset, the camera path traces out a golden spiral starting at the height corresponding to the originally sampled radius on the unit hemisphere. We use rendered scenes as observations, and the Cartesian coordinates of the camera and its velocities estimated through finite differences as the state. More details are provided in Appendix A.1.4.

## 5 Experiments and Results

Each model was trained on each of the 17 datasets described in Section 4 using the Adam [55] optimiser with learning rate $5 \times 10^{-4}$. For each model we performed a small hyperparameter search and reported the results from the best run. All of the models are implemented using the JAX framework [41] and run

| Dataset | HGN | | LGN NEW | | ODE | | ODE[TR] | | RGN Res | RGN | AR |
|---|---|---|---|---|---|---|---|---|---|---|---|
| | Forward | Backward | Forward | Backward | Forward | Backward | Forward | Backward | Forward | Forward | Forward |
| Mass-spring | **1.00(0.00)** | **1.00(0.00)** | **1.00(0.00)** | **1.00(0.00)** | **1.00(0.00)** | 0.00(0.00) | **1.00(0.00)** | 0.91(0.20) | **1.00(0.00)** | 0.59(0.11) | 0.20(0.40) |
| Mass-spring +c | 0.52(0.18) | **0.39(0.24)** | 0.00(0.00) | 0.00(0.00) | 0.52(0.23) | 0.00(0.00) | N/A(N/A) | N/A(N/A) | **0.56(0.21)** | 0.25(0.43) | 0.07(0.22) |
| Mass-spring +c +f | 0.41(0.17) | 0.13(0.05) | 0.00(0.00) | 0.00(0.00) | **0.98(0.08)** | 0.00(0.00) | 0.26(0.15) | **0.34(0.13)** | 0.94(0.11) | 0.21(0.06) | 0.15(0.36) |
| Pendulum | 0.11(0.02) | 0.03(0.01) | 0.15(0.03) | 0.15(0.03) | 0.26(0.22) | 0.00(0.00) | **0.45(0.32)** | **0.56(0.26)** | 0.13(0.03) | 0.00(0.00) | 0.00(0.00) |
| Pendulum +c | 0.08(0.03) | 0.01(0.00) | 0.10(0.04) | 0.08(0.04) | 0.10(0.10) | 0.00(0.00) | **0.12(0.08)** | **0.10(0.05)** | 0.10(0.06) | 0.05(0.02) | 0.00(0.01) |
| Pendulum +c +f | 0.13(0.06) | 0.03(0.02) | 0.13(0.08) | 0.27(0.13) | 0.16(0.11) | 0.00(0.00) | **0.30(0.33)** | **0.35(0.16)** | 0.19(0.18) | N/A(N/A) | 0.00(0.01) |
| D. pendulum | 0.13(0.04) | 0.01(0.01) | 0.16(0.06) | **0.17(0.06)** | 0.18(0.11) | 0.00(0.00) | **0.20(0.13)** | 0.15(0.08) | 0.19(0.10) | 0.10(0.05) | 0.00(0.01) |
| D. pendulum +c | 0.15(0.36) | 0.15(0.36) | 0.12(0.11) | 0.01(0.01) | 0.15(0.20) | 0.00(0.00) | **0.16(0.22)** | **0.26(0.32)** | 0.13(0.17) | 0.15(0.36) | 0.15(0.36) |
| D. pendulum +c +f | 0.11(0.09) | 0.04(0.02) | 0.09(0.08) | **0.56(0.21)** | **0.20(0.26)** | 0.00(0.00) | 0.17(0.18) | 0.42(0.15) | 0.18(0.27) | 0.01(0.05) | 0.01(0.05) |
| Two-body | 0.29(0.03) | 0.02(0.01) | 0.34(0.04) | 0.06(0.01) | **1.00(0.00)** | 0.00(0.00) | 0.99(0.05) | **0.94(0.13)** | 0.98(0.08) | 0.29(0.09) | 0.69(0.18) |
| Two-body +c | 0.21(0.05) | 0.06(0.04) | **0.25(0.11)** | 0.34(0.11) | 0.23(0.09) | 0.00(0.00) | 0.24(0.10) | **0.51(0.20)** | 0.24(0.10) | 0.19(0.10) | 0.15(0.09) |
| 3D room - spiral | 0.04(0.08) | 0.00(0.00) | 0.00(0.00) | 0.00(0.00) | 0.41(0.40) | 0.04(0.04) | **0.46(0.46)** | **0.45(0.47)** | 0.40(0.27) | N/A(N/A) | 0.00(0.00) |
| 3D room - circle | 0.07(0.14) | 0.00(0.01) | 0.08(0.20) | 0.08(0.20) | 0.32(0.31) | 0.04(0.05) | 0.35(0.43) | **0.34(0.43)** | **0.38(0.37)** | N/A(N/A) | 0.00(0.00) |
| MD - 4 particles | 0.18(0.07) | 0.01(0.01) | 0.19(0.12) | 0.23(0.12) | 0.28(0.05) | 0.06(0.01) | 0.26(0.11) | **0.28(0.11)** | **0.30(0.10)** | 0.00(0.00) | 0.04(0.01) |
| MD - 16 particles | 0.00(0.00) | **0.00(0.00)** | 0.00(0.00) | **0.00(0.00)** | 0.00(0.00) | **0.00(0.00)** | 0.00(0.00) | **0.00(0.00)** | 0.00(0.00) | 0.00(0.00) | **0.01(0.01)** |
| Matching pennies | **0.85(0.25)** | **0.86(0.25)** | **0.85(0.24)** | 0.76(0.29) | 0.54(0.33) | 0.00(0.00) | 0.76(0.28) | 0.83(0.24) | 0.40(0.24) | 0.55(0.34) | 0.20(0.40) |
| Rock-paper-scissors | 0.10(0.01) | 0.01(0.00) | **0.59(0.29)** | **0.67(0.27)** | 0.35(0.21) | 0.00(0.00) | 0.35(0.19) | 0.36(0.27) | 0.24(0.17) | 0.10(0.06) | 0.04(0.10) |
| Average rank | 4.47 | 2.59 | 4.18 | 2.12 | 2.88 | 3.47 | **2.47** | **1.41** | 3.0 | 5.65 | 5.65 |

Table 1: Forward and backward VPT scores for models with the best average VPT score per model class. VPT scores are presented in proportion to the maximal available dataset rollout length. In brackets are indicated the standard deviations of the scores across multiple trajectories.

on 4 P100 GPUs for 500,000 training steps with a total batch size of 128. It is important to mention that due to the matrix inversion required for computing the acceleration in the LGN model this model has a significantly larger computational cost per single step. In our setting we found it to be approximately five times slower than any of the other models, which could make it challenging for larger scale models.

**Evaluation metrics**  Evaluating how well a latent dynamical model has captured the true underlying system is still an open research problem. Because the model is trained directly on image observations, the only interpretable way of inspecting the time evolution of the learned system is through the observations. Typically in the literature train/test mean squared error (MSE) is used to evaluate models. However, in practice this has little reflection on the quality of the learnt dynamics. As an example in the mass-spring dataset, a model that has learned to produce a single black image, might have lower MSE compared to a model that has learned correct dynamics, but has inferred (wrongly) a phase with some fixed shift. For energy conservative systems it should be possible to roll out the dynamics forward or backward in time over very long time horizons if the underlying dynamics are learnt well. Hence, as an approximation for the model's ability to recover the underlying dynamics, we measure its ability to do such long-term extrapolation by measuring the *Valid Prediction Time* (VPT) (as per 56) using long trajectories of between 256 and 1000 steps depending on the dataset. VPT measures how long the model's trajectory remains close to the ground truth trajectory in the observation space. It corresponds to the first time step at which the model reconstruction significantly diverges from the ground truth:

$$\text{VPT} = \underset{t}{argmin} \left[ \text{MSE}(\boldsymbol{x}_t, \hat{\boldsymbol{x}}_t) > \epsilon \right] \tag{5}$$

where $\boldsymbol{x}_t$ is the ground truth, $\hat{\boldsymbol{x}}_t$ is the reconstructed observation at time $t$ and $\epsilon$ is a threshold parameter. Admittedly, even the VPT score is not perfect, as it still computes the divergence in image space, but in practice we found it to measure more accurately what we qualitatively have observed from sampled model's trajectories. For our datasets we found that a value of $0.025$ for $\epsilon$ worked well. All VPT scores are averaged over 20 trajectories and normalised by trajectory length. For completeness we also report in Appendix A.2 the "reconstruction" pixel mean squared error.

**Results**  The results for the best models per model class are summarised in Table 1. First, we can observe that across almost all datasets Neural ODE [TR] is on average the best model. Overall, this model class achieves the best results in 11/17 datasets for the forward rollouts and in 13/17 datasets for the backwards. It is closely followed by Neural ODE, and RGN Res when it comes to forward extrapolations (note that RGN Res is not capable of backward extrapolations by construction). In terms of backward extrapolations, HGN and LGN do well. RGN and standard autoregressive models performed poorly across the whole spectrum, suggesting that doing residual prediction (common to all other models) is important. Given that the datasets are sampled on a regular discrete grid, it is surprising to see the importance of modelling dynamics continuously that helped Neural ODE outperform RGN Res, which are identical apart from the discreteness of RGN Res.

Both the HGN and LGN models show good promise, especially in terms of getting good backward extrapolation performance effectively "for free", however, they are still somewhat lagging behind the Neural ODE [TR], which implements the same inductive biases of modeling continual dynamics,

predicting state update residuals and modelling the dynamics forward and backwards in time, but constrained in a different manner. It is unclear why the physics inspired models perform worse than the Neural ODE [TR] and we pose this as an open problem to the community. It is possible that these models require different optimization algorithms or architectures, but sifting through all such options is beyond the scope of this work. What makes HGN and LGN models an attractive option is that when they do learn a dataset well, their long extrapolations are consistently good, unlike those for the other models which have larger variances (shown in brackets in Table 1). The Lagrangian prior appears to have inferior performance to the Hamiltonian one on the majority of tasks, but not significantly, while being computationally more expensive and overall more unstable to train, thus providing the first fair head-to-head comparison between the two and indicating the superiority of the Hamiltonian prior.

VPT score close to 1 indicates that the dataset may be effectively "solved", whereby the model is consistently producing good reconstruction rollouts across 20 different inference seeds without diverging from the ground truth trajectory for 256-1000 steps while having been trained on 60. Hence, such models are likely to have recovered the underlying dynamics faithfully. It appears that mass-spring, mass-spring +f and two-body datasets were solved by at least one model, while MD-16 is the most challenging of the datasets, with no model being able to extrapolate beyond the training data sequences. The remaining 13 datasets form a nice continuum of difficulty to challenge progress in the field.

## 6    Conclusions and Limitations

In this paper we have reviewed the recently emerging literature that aims to bring ideas from classical mechanics as inductive biases to train models of dynamics from images. In particular, we have systematised the progressively larger body of literature to extract key themes on how to bring the Hamiltonian and Lagrangian formalism into the models while keeping the models as generally applicable as possible. We then performed a direct comparison between models with Hamiltonian and Lagrangian priors, Neural ODEs, Recurrent models and Autoregressive models on a set of 17 datasets of various complexity in terms of visuals and underlying dynamics. We have found that when it comes to modelling dynamics from images, the most promising general approaches are those that incorporate learning continuous dynamics with biases towards predicting both forward and backward in time and predicting residual state updates. Perhaps surprisingly, our results suggest that incorporating the Hamiltonian and Lagrangian formalism directly, however, is not yet as competitive to a standard Neural ODE [TR] that imposes the constraints from classical mechanics differently. That said, when models that do incorporate physical priors learn well, they have significantly less variance in their performance when extrapolating their dynamics over orders of magnitude longer time periods. In order to facilitate further progress in developing generally useful models of learning dynamics from high-dimensional observations, we release all datasets and model code.

The proposed benchmark is designed as a stepping stone toward the goal of developing general methods that can model natural physical dynamics from noisy and potentially partially observed high-dimensional observations. One consideration in the design of the benchmark was making it reasonably tractable to current approaches in order to serve as a sensitive measure of progress in the field. The downside to this consideration is that the benchmark does not capture the full set of challenges that must be faced to reach this ultimate goal – a set that is too expansive to incorporate immediately. For example, we do not consider the problem of reasoning in cluttered or visually complex scenes that arise in naturalistic, uncurated videos. Adding visual clutter to the scene is widely recognized to increase the difficulty of learning a task, even if the underlying task structure is kept fixed, e.g. [57, 58]. We also do not consider all dynamics of interest, including multi-body or contact dynamics that arise in robotics settings, or other, more challenging forms of dynamics such as non-rigid motion. As we have seen, the currently proposed datasets already present a challenge to a representative range of recently proposed models even without these additional complications. As such, we think that the proposed benchmark of 17 datasets serves as a good starting point by offering a good balance between tasks that are relatively easy but still have interesting dynamics, and tasks with harder to model dynamics and more complex visuals that are not yet solvable by current methods. We hope that progress on this benchmark will clear the ground for methods that can tackle more unconstrained dynamics and visual scenes, and we hope to extend the benchmark with the appropriate more challenging datasets in the future.

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
