# OpenReview forum: "Which priors matter? Benchmarking models for learning latent dynamics"
_NeurIPS.cc/2021/Track/Datasets_and_Benchmarks/Round1 — NeurIPS 2021 Datasets and Benchmarks Track (Round 1)_

### Official Review · Reviewer_RQVQ · 2021-07-04

**Rating:** 7
**Confidence:** 3
**Correctness:** Yes
**Clarity:** Yes

**Strengths:**

The paper studies an interesting direction, which is very relevant for model-based RL and robotics control community. The proposed datasets would be very useful for learning good dynamics models with physics priors. The implemented algorithms would also be helpful for the community to benchmark for future work. I think the contribution is significant enough to be accepted.

**Weaknesses:**

The authors didn't provide any simulated or real-world datasets in settings such as robotic manipulation and locomotion tasks, where effective model learning is important and methods with physics priors are unclear to work well. In general, the benchmark does not contain sufficient real-world datasets, which is a limitation.

**Additional Feedback:**

See above.

**Documentation:**

Yes

**Relation To Prior Work:**

Yes

**Summary And Contributions:**

This paper proposes a new benchmark for learning dynamics models leveraging physics priors from classic mechanics. The authors present 17 new datasets of dynamical systems ranging from real physical systems to camera movements in a 3D room and further implement and evaluate recently proposed algorithms with integrated physics prior in the 17 datasets.

---

### Official Review · Reviewer_dJJ5 · 2021-07-06
**Nice suite of tasks and surprising results**

**Rating:** 8
**Confidence:** 4

**Strengths:**

1. This work investigates large body of recent method that uses physics prior in latent dynamics model;
2. The purposed datasets include various complexities of visual dynamics that can be seen as a standard test set of evaluating dynamics model ;


**Weaknesses:**

1. All the environments are artificial and the visual features fairly simple. I think it could be also useful to adding an extra dataset with more realistic visuals.
2. While the results shows a nice comparison among different benchmarks, more analysis can be done. It is also interesting to see how physics priors affect the quality of learned latent variables and what actually limit the performance of these model.
3. I think RGN Res and RGN can also be trained in the same manner as Neural ODE [TR] (i.e. training forward and backward in time). I think this can provide more comprehensive comparison across these model.

**Additional Feedback:**

I think more analysis can be done. One worthy investigation can be what kind of task is more suitable to models with physics prior. In general, I think this is a paper to accept.

**Clarity:**

The paper is well-written and the design of the dataset are well-explained. The comparison of benchmark methods shows clear results of how each model perform of the tasks.

**Correctness:**

I think this dataset is built in a sound way and the evaluation methods is designed appropriately.

**Documentation:**

The design of each environments are well explained. The authors provides codes for models and datasets. I did not run them but I think they should be sufficient to reproduce results.

**Ethics:**

No, I don't have any ethical concerns.

**Relation To Prior Work:**

Yes, this paper aims to evaluate methods from previous works in a standard way.

**Summary And Contributions:**

This paper presents a suite of datasets that are generated by physical systems to investigate the effect of integrating physics prior into deep neural network for learning latent dynamics from visual observations. Specifically, the authors study the effect of  using Hamiltonian and Lagrangian inductive bias while learning dynamics of various physics system. The datasets purposed by the authors include toy physics problems, non-transitive zero-sum games, molecular dynamics, and 3D environments. The authors implement previously purposed model with physics dynamics and surprisingly, the experimental results shows that a baseline, the Neural ODE, achieves the best results among all models. This suggests that both method of employing physics methods is not more useful than Neural ODE.

---

### Official Review · Reviewer_CJoj · 2021-07-06
**A Well-written overview and evaluation of learning latent dynamics**

**Rating:** 8
**Confidence:** 3
**Correctness:** Yes.
**Clarity:** Yes, it is very clear and well-writte…

**Strengths:**

The paper is well-written, and a good survey of these methods. The results section is clear and well written, and the experiments are otherwise clear and exhaustive. I think the paper has a valuable contribution to the community for researching these kinds of models.

**Weaknesses:**

I believe for these constructed datasets one should have access to underlying physical dynamics of the system, so I'm not sure why the HGN and LGN models were not evaluated in terms of the distance between the predicted and ground truth kinetic and potential energy. I think this evaluation could be useful for comparison different methods in HGN and LGN models, and it would be useful for the datasets to provide ground truth physics information, such as the kinetic and potential energy of particles in the system.

**Additional Feedback:**

N/AA

**Documentation:**

I didn't see information about a URL, availability and maintenance of the dataset, but these may be in the supplementary materials.

**Ethics:**

No.

**Relation To Prior Work:**

Yes.

**Summary And Contributions:**

The paper provides a clear and well-written overview of methods for learning latent dynamics with physics-based priors. The paper does a good job of clearly categorizing the differences between prior methods, and is a good resource as a survey paper of these methods.

I think the authors could include more explanation of the multi-agent cyclic games portion of the dataset in the main paper, rather than in the appendix. The main text could include some description of "continuous-time multi-population replicator dynamics", a term which I believe isn't well-known in the machine learning community.

---

### Note · ~Aleksandar_Botev1 · 2021-11-08

https://console.cloud.google.com/storage/browser/dm-hamiltonian-dynamics-suite

---

### Decision · Program_Chairs · 2021-07-26

**Decision:**

Accept

**Comment:**

The paper presents a new suite of physics-based benchmarks for comparing the performance of neural network models which learn latent physical dynamics from visual input. All reviewers praised the proposed datasets as well as the controlled comparison of prior approaches on this new dataset. The main weakness identified by reviewers is the lack of a dataset with more realistic visuals; however the authors were able to address this and other more minor concerns in their responses and revised paper. In the end all authors voted to accept the paper. Congratulations on having your paper accepted to the NeurIPS 2021 Track on Datasets and Benchmarks! The authors are encouraged to take the feedback from reviewers into account when preparing the final version of their paper.